# Evaluation of the Impacts of Climate Change on Sediment Yield from the Logiya Watershed, Lower Awash Basin, Ethiopia

**Nura Boru Jilo [1],\***, **Bogale Gebremariam [2]**, **Arus Edo Harka [1]**, **Gezahegn Weldu Woldemariam [3]** 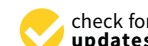 **and Fiseha Behulu [4]**

1. Hydraulic and Water Resources Engineering Department, Haramaya Institute of Technology (HiT), Haramaya University, P.O. Box 138 Dire Dawa, Ethiopia
2. Department of Hydraulic and Water Resources Engineering, Arba Minch Water Technology Institute, Arba Minch University, P.O. Box 21 Arba Minch, Ethiopia
3. Geoinformation Science Program, School of Geography and Environmental Studies, Haramaya University, P.O. Box 138 Dire Dawa, Ethiopia
4. Institute of Technology (AAiT), Addis Ababa University, Addis Ababa 385, Ethiopia
* Correspondence: nuraboru1@gmail.com; Tel.: +251-(0)-9-10-08-97-39

**Abstract:** It is anticipated that climate change will impact sediment yield in watersheds. The purpose of this study was to investigate the impacts of climate change on sediment yield from the Logiya watershed in the lower Awash Basin, Ethiopia. Here, we used the coordinated regional climate downscaling experiment (CORDEX)-Africa data outputs of Hadley Global Environment Model 2-Earth System (HadGEM2-ES) under representative concentration pathway (RCP) scenarios (RCP4.5 and RCP8.5). Future scenarios of climate change were analyzed in two-time frames: 2020–2049 (2030s) and 2050–2079 (2060s). Both time frames were analyzed using both RCP scenarios from the baseline period (1971–2000). A Soil and Water Assessment Tool (SWAT) model was constructed to simulate the hydrological and the sedimentological responses to climate change. The model performance was calibrated and validated using the coefficient of determination ($R^2$), Nash–Sutcliffe efficiency (NSE), and percent bias (PBIAS). The results of the calibration and the validation of the sediment yield $R^2$, NSE, and PBIAS were 0.83, 0.79, and −23.4 and 0.85, 0.76, and −25.0, respectively. The results of downscaled precipitation, temperature, and estimated evapotranspiration increased in both emission scenarios. These climate variable increments were expected to result in intensifications in the mean annual sediment yield of 4.42% and 8.08% for RCP4.5 and 7.19% and 10.79% for RCP8.5 by the 2030s and the 2060s, respectively.

**Keywords:** Logiya watershed; climate change; CORDEX-Africa; SWAT; RCP; sediment yield

## 1. Introduction

Climate change is becoming a major environmental concern because increasing scientific evidence shows the high concentration of greenhouse gases (GHGs) in the atmosphere, and frequent hydro-meteorological extreme events are becoming the 21st century phenomenon [1,2]. Climate change results in significant impacts on life and natural resources [3]. The consequences of climate change involve adverse impacts on environment, hydrological cycle, water resources, agriculture and food security, human health, terrestrial ecosystems, and biodiversity [3–8]. These events are mostly manifested through subsequent influences of climate variables, including precipitation, temperature, and evapotranspiration [9–11]. Despite variations across regions, heat waves and extreme precipitation events will become more intense and more frequent. It is also estimated that the mean annual global

air temperature induced by GHG emissions will likely increase by 1.4–2.0 °C by the end of the 21st century [10]. The rising GHG concentration due to continued gas emissions from different sources to the atmosphere affects climate variables and consequently alters hydrological cycles [10,12–15]. The change in climate—mainly due to precipitation and temperature patterns—could significantly influence soil erosion rates, streamflow, and sediment yield, which (directly or indirectly) adversely affects water resource availability and ecosystems [13,16–19]. In connection to this, numerous studies have indicated that changes in precipitation, temperature, and the interactions of these with land use and land cover change (LULCC) will be the main climate change related stress expected to exacerbate soil loss and sediment transport [20–23].

Soil erosion is a naturally occurring phenomenon through which the most productive topsoil materials are detached, transported, and deposited downstream by wind, water, and gravitational forces [24]. The displacement of soil layers due to water induced forces is causing continuous erosion, which in turn is changing the Earth's surface [16,25]. This is increasing over recent years due to climate change and anthropogenic factors, resulting both in on-site and off-site effects [16,24]. Due to all these factors, soil erosion rates and the amount of suspended sediment discharges are being exacerbated across the world.

Over the past decade, the impacts of climate change on sediment yields have been widely studied around the globe [13,19,25–28]. Also, studies show that sediment transport is highly influenced by extreme precipitation and river discharge due to the non-linear relationship between water discharge and sediment transportation rates [29]. Moreover, Francipane et al. [25] reported that the magnitudes of mean and extreme events of sediment yield are expected to decrease with a high probability in the Walnut Gulch Experimental watershed in southeastern Arizona, USA. However, on their part, Zhang et al. [19] indicated that climate change is expected to increase the annual average sediment yield by 4–32% when compared to the base period for the Zhenjiangguan watershed, China. Azim et al. [12] examined the impact of climate change on the sediment yield of the Naran watershed, Pakistan using global climate model (GCM) predictors from the Special Report on Emission Scenarios (SRES) A2 and B2 and found mean annual sediment yield increases of 5–6% and 9–11% under the A2 and the B2 scenarios for time horizons of 2011–2040 and 2041–2070, respectively. Likewise, Shrestha et al. [12] used the Soil and Water Assessment Tool (SWAT) model in the Nam Ou Basin in northern Laos to predict the impact of climate change on sediment yield and reported changes in an annual sediment yield ranging from a 27% decrease to about a 160% increase. Further, Azari et al., [27] conducted a similar study focusing on the Gorganroud River Basin in northern Iran and found that the annual sediment yield would increase by 47.7%, 44.5%, and 35.9% for the 2040–2069 time series of the A1F1, the A2, and the B1 emission scenarios, respectively.

Farmers in the Horn of Africa, including Ethiopia, are frequently vulnerable to climate change extremes such as floods and droughts, both of which are the most common hydro-meteorological extremes in tropical countries [5–8,30–32]. Ethiopia is a country in which the economy largely depends on rain-fed agriculture, which is highly vulnerable to the impacts of climate change, mainly changes in precipitation patterns [33,34]. Precipitation pattern and character in Ethiopia are predominantly controlled by the oscillation of the Intertropical Convergence Zone. This precipitation has a strong impact on soil erosion. A change in precipitation intensity and seasonal distribution results in a significant impact on the soil erosion rate [20]. According to Zhang and Nearing [35], precipitation and soil loss are directly proportional to each other.

The Logiya watershed is one of the sub-watersheds in the Lower Awash River Basin (LARB). It is located in the arid lowlands of the Afar Region in the northeastern part of Ethiopia. The upstream of the Logiya Watershed extends up to North Wollo Zone of the Amhara Regional State, and the outlet of the watershed is situated in the Afar Regional State. The precipitation and the temperature of the watershed both fluctuate, which subsequently results in a variation of the river flow in the watershed under undue climate change conditions. The Logiya watershed is characterized by severe

land degradation and desertification, but high flood also occurs, mostly in the upstream areas during high rain seasons [36].

Several studies have been conducted to assess flooding and droughts caused by climate change in the LARB, including the Logiya watershed [36–42]. However, none of these studies addressed the climate change impacts on sediment yield both in the LARB and the Logiya watershed. The magnitude of climate change effects on sediment yield varies depending on the region in focus and the climate scenario taken into consideration [43]. Usually, heterogeneity of the climate conditions is wrongly assumed in the previous scenarios by some researchers. However, it is necessary to regionalize the assessment of climate change impacts. In other words, understanding the peculiar effects of climate change on sediment load is critical for watershed management [44].

In order to examine the non-linear interplays between climate-induced processes that influence sediment discharge at the watershed level, hydrological and sediment models can be used in combination with climate projections from global circulation models and regional climate models [25,45]. The present study used representative concentration pathways (RCPs), which supersede the Special Report on Emission Scenario (SRES), a newly developed scenario that provides input for climate models and a distributed hydrological model, SWAT, to estimate potential impacts of climate change on sediment yield in the Lower Awash Logiya watershed. The specific objectives of the study were to (i) assess future precipitation, air temperature, and evapotranspiration; (ii) estimate the spatial variation of sediment yield under different climate scenarios; and (iii) evaluate the impacts of climate change on sediment yield from the Logiya watershed under RCPs scenarios.

## 2. Materials and Methods

### 2.1. Description of the Study Area

The Awash River Basin (ARB) is one of the major river basins in Ethiopia. It originates in the highlands of Central Ethiopia and flows northeastwards, where it eventually drains into the Lake Abe. The total length of the river is about 1200 km and has a total drainage area of about 110,000 km$^2$. The ARB is divided into three distinct zones: Upper Basin, Middle Basin, and Lower Basin. The Logiya watershed is among the sub-watersheds of LARB and is situated in the western part of the basin between 11°28'21'' N and 12°04'55'' N and 39°40'30'' E and 40°56'15'' E (Figure 1). The watershed drainage area is about 3151.86 km$^2$ up to the outlet. The elevation of the study area ranges from 3426 to 379 m a.m.s.l. with an average altitude of 890.6 m. Awash River Basin ranges from cold high mountainous zones to semi-desert lowlands with extreme ranges of temperature and precipitation [40]. There are three seasons in the Awash River Basin based on the movement of the Intertropical Convergence Zone (ITCZ) and the amount of precipitation timing. The three seasons are named: Kiremt, which is the main rainy season (June–September), Bega, which is the dry season (October–January), and Belg, the small rainy season (February–May). Therefore, the precipitation was categorized under bi-modal pattern. The mean annual precipitation varies from 1600 mm in the elevated areas to 160 mm in the LARB. In the same way, the mean annual temperature of Awash River Basin ranges from 20.8 °C in the upper part to 29 °C in the lower part. Figure 2 shows observed mean monthly precipitation as well as maximum and minimum temperatures of the Logiya watershed.

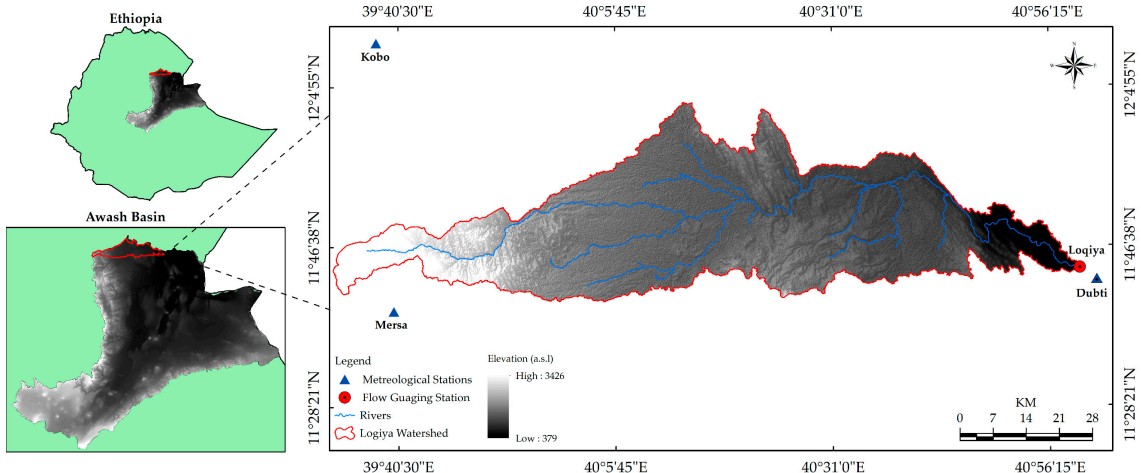

**Figure 1.** Location of the Logiya watershed, meteorological and flow gauging stations.

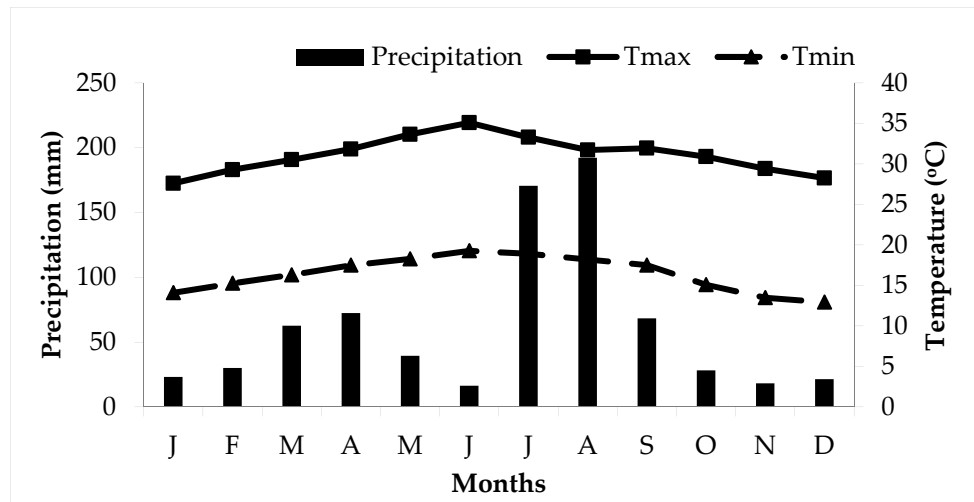

**Figure 2.** Mean monthly precipitation and temperature (Tmax and Tmin) selected stations in the study area (1988–2016).

## 2.2. Data Sources

The required input data used to run the SWAT models necessary for this study were drawn from hydrological and meteorological data, soil maps, Land Use/Land Cover (LULC) maps, and digital elevation models (DEM) of the watershed. These were collected from different sources, to be discussed below.

### 2.2.1. Digital Elevation Model

The DEM is used to describe the elevation of any point in a given area at a specific spatial resolution. The 30 m × 30 m resolution DEM data covering the watershed area was downloaded from Advanced Spaceborne Thermal Emission and Reflection Radiometer (ASTER) Global Digital Elevation Model (GDEM) website [46]. It was required for the SWAT model to delineate the watershed, the sub-watershed, the topography, and the drainage pattern of the study area. The DEM data was loaded into Arc SWAT interface after being re-projected into the World Geodetic System (WGS 84) spheroid, a spatial reference system of the Universal Transverse Mercator (UTM) with Datum Zone 37 N.

2.2.2. Hydrological and Meteorological Data

This includes the daily observed climate data from 1988–2016 comprising data about the precipitation, maximum and minimum temperature, solar radiation, wind speed, and relative humidity. These were collected from the National Meteorological Agency (NMA) of Ethiopia [47]. For this purpose, four stations, namely, Dubti, Mersa, Kobbo, and Bati (Table 1), were selected based on the quality and the availability of long year recorded data. Moreover, hydrological data including streamflow and suspended sediment yield at Logiya gauging station was collected from the Ministry of Water, Irrigation, and Electricity (MoWIE). Daily streamflow data at the outlet of the watershed were used for the calibration and the validation of the SWAT model. The missing hydrological data were filled by linear regression from the nearby Mille river gauged station. Moreover, the meteorological data of each neighboring station were filled by the Normal-Ratio Method, while the homogeneity and the consistency of data were tested and used as inputs for the climate and the hydrological models [48–50].

**Table 1.** Meteorological stations near the Logiya watershed and percentage of missing data.

| Stations | Latitude | Longitude | Elevation (m) | Missing (%) |
|---|---|---|---|---|
| Dubti | 11.72 | 41.01 | 376 | 6.62 |
| Mersa | 12.13 | 39.63 | 1470 | 7.72 |
| Kobbo | 11.66 | 39.67 | 1578 | 14.75 |
| Bati | 11.2 | 40.02 | 1660 | 4.02 |

2.2.3. Soil Data

The soil data of the Logiya watershed was obtained from the MoWIE. This data was prepared according to the Food and Agriculture Organization (FAO) classification scheme. Accordingly, there were ten soil types found in the study area (Figure 3a). These are included Calcaric flubisols, Chromic luvisols, Dystric nitisols, Eutric cambisols, Eutric, regosols, Haplic xerosols, Leptosols, Orthic solonchaks, Vertic cambisols, and Vitric andosols, with a proportion of each class contributing 29.04%, 0.15%, 0.90%, 2.72%, 19.16%, 6.57%, 12.22%, 18.76%, 3.27%, and 7.21%, respectively, to the total watershed area.

2.2.4. Land Use/Land Cover

The most dominant LULC classes in the Logiya watershed included grass steppe, shrub, tree steppe and bare soil and very sparse vegetation (Figure 3b). It has been found that LULC has impacts on climate change, which in turn influences surface runoff, soil erosion, sediment load, and evapotranspiration in a watershed as the SWAT model simulation shows [51].

2.2.5. Climate Models and RCPs Scenarios

The coordinated regional climate downscaling experiment (CORDEX) is a program sponsored by the World Climate Research Program (WCRP) to generate historical and future climate projections at the regional scale. A regional climate model (RCM) was used to advance the predictive model of the Earth's climate in the scientific analysis of the dominant sets of the governing processes. It describes climate change on a regional scale [52]. Thus, the RCM was integrated into the CORDEX-Africa domain with a horizontal grid resolution of $0.44° × 0.44°$ (50 km × 50 km) [53].

The historical simulations were forced by observed data and anthropogenic atmospheric composition covering the period 1951–2005, whereas the projected scenarios between 2006 and 2100 were derived through RCP4.5 and RCP8.5 scenarios. Also, simulated data of RCP4.5 and RCP8.5 scenarios were generated from Hadley Global Environment Model 2-Earth System (HadGEM2-ES) of the global climate model outputs. These were dynamically downscaled from the CORDEX-Africa database.

Furthermore, future scenario analyses RCP4.5 (medium emission scenarios) and RCP8.5 (high emission scenarios) were used to assess the precipitation, maximum and minimum temperature, evapotranspiration, and sediment yield. The analyses were performed in two-time horizons in a time period of 30 years for both baseline and future periods. The baseline period used for this study was 1971–2000, while future scenario analysis involved 2020–2049 (2030s) and 2050–2079 (2060s).

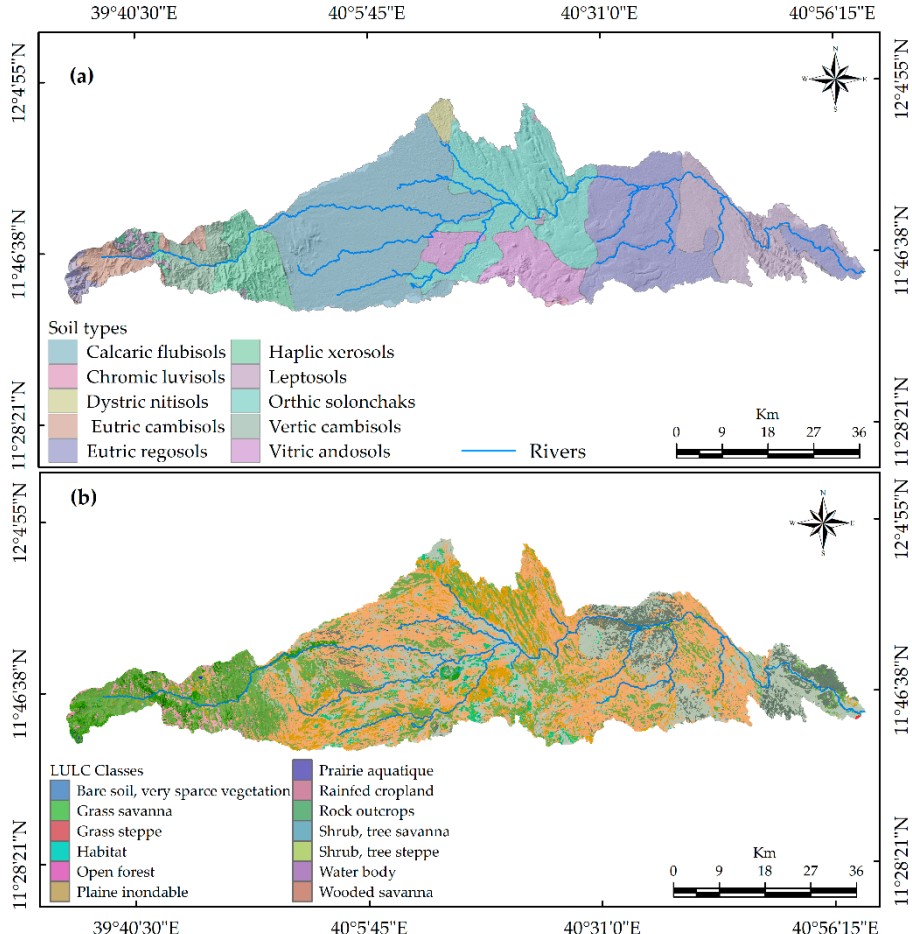

**Figure 3.** Soil types (**a**) and Land Use/Land Cover (LULC) classes (**b**) in Logiya watershed.

### 2.2.6. Accuracy of Precipitation Simulations from the Climate Model

The precipitation simulations from the HadGEM2-ES global climate model outputs were evaluated using observed data. Accuracy of the precipitation was evaluated using statistical methods, including bias, correlation coefficient (CC), and coefficient of variation (CV), computed as follows [54]:

$$Bias = \frac{\overline{R}_{rcm} - \overline{R}_{obs}}{\overline{R}_{obs}} \times 100 \tag{1}$$

$$CV = \frac{\sigma R}{\overline{R}} \times 100 \tag{2}$$

$$Correl = \frac{\sum_{i=1}^{N} (R_{obs} - \overline{R}_{obs})(R_{rcm} - \overline{R}_{rcm})}{\sqrt{\sum_{i=1}^{N} (R_{obs} - \overline{R}_{obs})^2 (R_{rcm} - \overline{R}_{rcm})^2}} \tag{3}$$

where $\overline{R}$ is an average precipitation over the watershed; *rcm* and *obs* are subscripts representing the precipitation amount over the watershed either from RCM simulation or observed datasets,

respectively; σ indicates the standard deviation of either the RCM or the observed precipitation data, and R represents estimated statistics individually either for RCM or observed precipitation amount.

### 2.2.7. Bias Correction Method

Using downscaled regional climate data for impact assessment without any bias correction may lead to considerable deviation when a hydrologic model is forced with a biased RCM [10,55–58]. Hence, a bias correction method was applied for each daily precipitation and temperature dataset derived from CORDEX-Africa to minimize the systematic statistical deviation of climate input data from the observational one before using them for any modeling and future climate change projection. The bias correction method used for this study was non-linear correction to adjust observed and simulated precipitation data. Correction factors were computed from the statistics of the observed and the simulated variables. The principle of this method shows the mean and the standard deviation of the daily precipitation data becoming equal to those of the observed data [59–61].

$$P^* = aP^b \tag{4}$$

where $P^*$ is the corrected value of precipitation, and $a$ and $b$ are power transformation parameters determined for the period of a year, which were obtained from calibration in the baseline period and subsequently applied to the projection period.

Hence, the bias correction of temperature only involved shifting and scaling to adjust the mean and the variance [62,63]. The corrected daily temperature $T_{corr}$ was obtained as:

$$T_{corr} = \overline{T}_{obs} + \frac{\sigma(T_{obs})}{\sigma(T_{rcm})}\left(T_{rcm} - \overline{T}_{obs}\right) + \left(\overline{T}_{obs} - \overline{T}_{rcm}\right) \tag{5}$$

where $T_{corr}$ is the corrected daily temperature; $\overline{T}_{rcm}$ is the uncorrected daily temperature from the RCM model; $\overline{T}_{obs}$ is observed daily temperature; $\overline{T}_{obs}$ and $\overline{T}_{rcm}$ are mean temperatures for observed and simulated datasets, respectively.

### 2.2.8. Statistical Test for Trend and Variability Analysis

For this study, non-parametric statistical method, the Mann-Kendall (MK) trend test, was used to identify the presence of trends in the time series at the 5% significance level [64–67]. The MK trend test analysis was recommended by the World Meteorological Organization (WMO) and has been widely used in practice to evaluate the significance of monotonic trends in hydrological and meteorological time series [14,68,69] and to determine the changes (increasing, decreasing, or trendless) in the values of climatic variables.

For a given time series $X (x_1, x_2, \ldots, x_n)$, the statistic $S$ is defined as:

$$S = \sum_{i=1}^{n} \sum_{j=i+1}^{n-1} sgn\left(x_j - x_i\right) \tag{6}$$

where

$$Sign\left(X_j - X_i\right) \begin{cases} 1, & x_j > x_i, \\ 0, & x_j = x_i, \\ -1, & x_i < x_j, \end{cases} \tag{7}$$

$x_i$ and $x_j$ are the data values in the time series, and $n$ is the number of data points in the time series,

The variance S is calculated using the following equation when the value of $n \geq 8$;

$$Var(s) = \frac{n(n-1)(2n+s)}{18} \tag{8}$$

The standard normal test statistic is calculated using the following equation;

$$Z = \begin{cases} \frac{(1-S)}{\sqrt{Var(S)}}, & S > 0, \\ 0, & S = 0, \\ \frac{(1+S)}{\sqrt{Var(S)}}, & S < 0. \end{cases} \tag{9}$$

The MK statistic is denoted by *S* follows the standard normal distribution. Positive and negative values of *S* denote an upward and downward trends, respectively. The null hypothesis of no trend is rejected if $|S| > 1.96$ at the 5% significance level.

### 2.3. Sediment Rating Curve Development

Sediment is some of the most important data in the hydrological study for the estimate of sediment transport. Due to the scarcity of continuous sediment data, the sediment rating curve was necessary to develop the relationship between daily streamflow and sediment data measured at the outlet of the Logiya watershed. It displayed the rate of sediment transport as a function of streamflow [26]. The suspended load in milligram per liter was converted to tons/day as follows:

$$Q_s = 0.0864 \times Q \times C \tag{10}$$

where $Q_s$ is sediment load in (t day$^{-1}$), $Q$ is streamflow (m$^3$ s$^{-1}$) and C is suspended sediment concentration (mg L$^{-1}$). As depicted in Figure 4, the sediment rating curve was developed by using linear regression techniques, and the curve is expressed in the form of a power-law type equation [29,70–72].

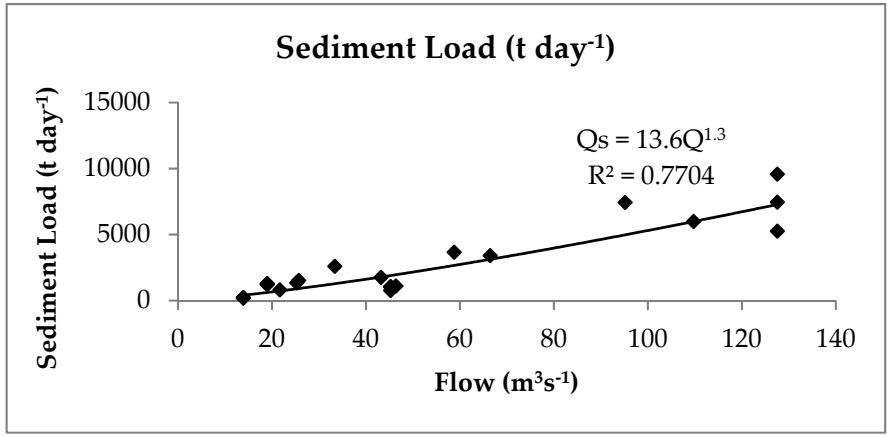

**Figure 4.** Sediment rating curve of the Logiya River at the Logiya gauge station.

### 2.4. SWAT Model Setup

### 2.4.1. SWAT Model Description

SWAT is one of the most powerful hydrologic models [73]. It is physically semi-distributed, conceptually and computationally efficient and operates on a daily time step according to the soil properties, topography, vegetation and land management practices in the watershed [51]. The model was developed by the United States Department of Agricultural Research Service [74] to predict and qualify the impact of agricultural management practices on water, sediment and chemical yields in different size of catchments with varying soils, land use and management conditions over long periods of time [51,75].

The model has been widely applied to the different sizes of the watersheds and its applications all over the water resources, including the effects of climate and soil erosion [51]. In our study, the SWAT

model was used to estimate hydrologic components such as surface runoff, evapotranspiration, groundwater flow, sediment yield and for each hydrologic response unit (HRU). In the hydrological component, surface runoff was estimated separately for each sub-basin of the total watershed area and routed to obtain the total surface runoff for the watershed. Surface runoff volume was estimated from daily rainfall using modified soil conservation service curve number (SCS-CN) and Green and Ampt methods [76]. Potential evapotranspiration can be estimated by three methods: the Hargreaves [77], the Priestley–Taylor [78], and the Penman–Monteith [79]. Based on the availability of observed data, the Hargreaves method was adopted for this study. In each HRU sediment yield was estimated using a modified universal soil loss equation (MUSLE). The modified universal soil loss equation was determined as [74]:

$$Sed = 11.8 \times \left(Q_{surf} \times q_{peak} \times Area_{HRU}\right)^2 \times K_{USLE} \times C_{USLE} \times P_{USLE} \times LS_{USLE} \times CFRG \qquad (11)$$

where *Sed* is the sediment yield (metric ton day$^{-1}$), $Q_{surf}$ is the surface runoff volume (mm ha$^{-1}$), $q_{peak}$ is the peak runoff rate in m$^3$ s$^{-1}$, $Area_{HRU}$ is the area of HRU in ha, $K_{USLE}$ is the soil erodibility factor, $C_{USLE}$ is the cover and management factor, $P_{USLE}$ is the support practice factor, LS is topographic factor, and CFRG is the course fragment factor.

The routing phase of SWAT defines the movement of water, nutrients, sediment, and pesticides through the channel network of the watershed into the outlet. The sediment-routing model that simulates sediment transport in the channel network consists of two components operating simultaneously—deposition and degradation [74]. The amounts of deposition and degradation are based on the maximum concentration of sediment in the reach and the concentration of sediment in the reach at the beginning of the time step.

The final amount of sediment in the reach was determined by:

$$Sed_{ch} = Sed_{ch,i} - Sed_{dep} + Sed_{deg} \qquad (12)$$

where $Sed_{ch}$ is the amount of suspended sediment in the reach (metric tons days$^{-1}$), $Sed_{ch,i}$ is the amount of suspended sediment in the reach at the beginning of the time period (metric tons days$^{-1}$) $Sed_{dep}$ is the amount of sediment deposited in the reach segment (metric tons days$^{-1}$) and $Sed_{deg}$ is the amount of sediment re-entrained in the reach segment (metric tons days$^{-1}$). The amount of sediment transported out of the reach was calculated as:

$$Sed_{out} = Sed_{ch} \times \frac{V_{out}}{V_{ch}} \qquad (13)$$

where $Sed_{out}$ is the amount of sediment transported out of the reach (metric tons days$^{-1}$), $V_{out}$ is the volume of outflow during the time step (m$^3$), and $V_{ch}$ is the volume of water in the reach segment (m$^3$).

### 2.4.2. Watershed Delineation

The watershed delineator tool in ArcSWAT 2012 interface allowed us to delineate the watershed and the sub-watersheds using the 30 × 30 m DEM. Flow direction and accumulation are the concepts behind the definition of the stream network of the DEM in SWAT. The monitoring point was added manually, and the numbers of the sub-basin were adjusted accordingly. There were 33 sub-catchments with an area ranging between 239.71 and 47,022.06 hectares.

### 2.4.3. Hydrologic Response Units (HRU) Analysis

HRU analysis helps to load land use and soil maps as well as to incorporate classification of HRU into different slope classes. HRUs were assigned to each sub-catchment based on a 10% threshold value for LULC, soil, and slope categories, as suggested by the user's manual of SWAT to increase the chance of inclusion.

Based on the topographical characteristics of the terrain in the DEM, the SWAT model described watershed spatial variability by further splitting the sub-watershed into homogeneous characteristics, lumped land areas, and HRUs based on topography, soil, land use, and slope. Multiple land use/soil/slope methods were used to define the HRU with land use (10%), soil (10%), and slope (15%) thresholds. The number of HRUs was defined by eliminating the percentages of land use, soil, and slope values that covered a percentage of the sub-catchment area less than the threshold level. Accordingly, the Logiya watershed was divided into a total of 234 HRUs of different LULC, soil, and slope combinations.

## 2.5. SWAT Calibration and Uncertainty Procedures

SWAT Calibration and Uncertainty Procedures (SWAT-CUP) is an interface that was developed for the SWAT model [80]. It is a computer program for integrated various sensitivity analysis, calibration, and validation programs for the SWAT model. Sensitivity analysis is used to check the rate of change in output parameters with respect to the input parameters of the model. For this study, the measured stream flow and sediment yield data obtained from the discharge–sediment rating curve at the outlet of the Logiya watershed were used to calibrate and validate the model by SWAT-CUP using Sequential Uncertainty Fitting version-2 (SUFI-2) algorithm from 1990 to 2005. The first two years were considered as the model warm-up period, the period from 1992 to 1999 was used for calibration, while the remaining data (2000–2005) were used for model validation.

The model performance efficiency was determined by comparing observed against simulated hydrographs. The coefficient of determination ($R^2$), the Nash–Sutcliffe simulation efficiency (NSE), and the percent bias (PBIAS) were used to evaluate the model's performance during calibration and validation processes to test the goodness of fit between monthly simulated and observed values. These parameters were important to investigate and determine the accuracy of the SWAT model simulations. The determination coefficient ($R^2$) describes the proportion of variance in measured data from the model. The value of $R^2$ ranged from zero (which indicated the model was poor) to one (which indicated the model was perfect), with higher values indicating less error variance; typical values greater than 0.6 were considered acceptable [53]. The $R^2$ was calculated using the following Equation:

$$R^2 = \frac{\left[\sum_{i=0}^{n}\left(Q_{Obs} - \overline{Q}_{Obs}\right) \times \left(Q_{Sim} - \overline{Q}_{Sim}\right)\right]^2}{\sum_{i=0}^{n}\left(Q_{Obs} - \overline{Q}_{Obs}\right)^2 \sum_{i=0}^{n}\left(Q_{Sim} - \overline{Q}_{Sim}\right)^2} \tag{14}$$

If, $R^2 = 0$, none of the variances in the measured data were replicated by the model predictions and if $R^2 = 1$, it indicated that all of the variances in the measured data were replicated by the model predictions. NSE indicated how well the plots of the observed versus the simulated data fit the 1:1 line. NSE was computed as follows (Equation (15)):

$$NSE = 1 - \frac{\sum_{i=0}^{n}(Q_{Obs} - Q_{Sim})^2}{\sum_{i=0}^{n}\left(Q_{Obs} - \overline{Q}_{Obs}\right)^2} \tag{15}$$

The value of NSE ranged from negative infinity to one (perfect) where values lower than zero indicated that the mean observed value was a better predictor than the simulated value and revealed unacceptable model performances. NSE values greater than 0.5 showed the simulated value was a better predictor than the mean measured value and was generally viewed as acceptable performance [81].

PBIAS was used to measure the average tendency of the simulated data to be larger or smaller than the observed values. PBIAS is expressed in percentage; a lower absolute value of the PBIAS is better and determines model performance.

$$PBIAS = \frac{\sum_{i=0}^{n}(Q_{Obs} - Q_{Sim})}{\sum_{i=0}^{n}(Q_{Obs})} \times 100 \tag{16}$$

where n is the number of observations during the simulation period, $Q_{Obs}$ is the observed flow data, $Q_{Sim}$ is the simulated flow value with the respective time, and $Q_{Obs}$ and $Q_{Sim}$ are the arithmetic means of the observed and the simulated values, respectively

The optimal value of PBIAS was zero, with low-magnitude values indicating accurate model simulation. Positive values indicated model underestimation bias, and negative values indicated model overestimation bias [82].

## 3. Results and Discussion

### 3.1. Statistical Trend Analysis of Historical Climate Variability

The variation trends of annual precipitation, maximum and minimum temperature, and sediment yield in the Logiya watershed from 1971–2000 can be found in Figure 5a–d. In the watershed, the annual precipitation and the sediment yield showed insignificant decreasing trends, and the annual maximum and minimum temperatures showed insignificant increasing trends, which meant the p-value was greater than the significance level α (5%) (Table 2). Therefore, results of mean annual precipitation as well as maximum and minimum temperatures were related to other findings reported for the country. For example, Atlas [83] reported decreasing precipitation and increasing temperature trends in the Afar region during the period 1981–2014. Likewise, Jury and Funk [84] showed annual air temperature increased and precipitation decreased over the time series 1948–2006 in Ethiopia.

The mean, the maximum (Max), the minimum (Min), the standard deviation (SD), and the coefficient of variance (CV) of annual precipitation, maximum and minimum temperature, and sediment yield of the watershed are depicted in Table 2. The mean annual precipitation of the watershed during the baseline period was 125.4 mm with 54.3 mm standard deviation and 43.3% CV. The minimum and the maximum ever-recorded precipitations were 58.0 mm and 304.5 mm, respectively. CV was used to classify the degree of variability in the historical data events as less (CV < 20), moderate (20 < CV < 30), high (CV > 30), or very high (CV > 40%), but CV > 70% indicated extremely high inter-annual variability [69]. The coefficients of variation of annual precipitation and sediment yield were above 40% and 70%. This indicated very high variability of precipitation and sediment yield over the watershed.

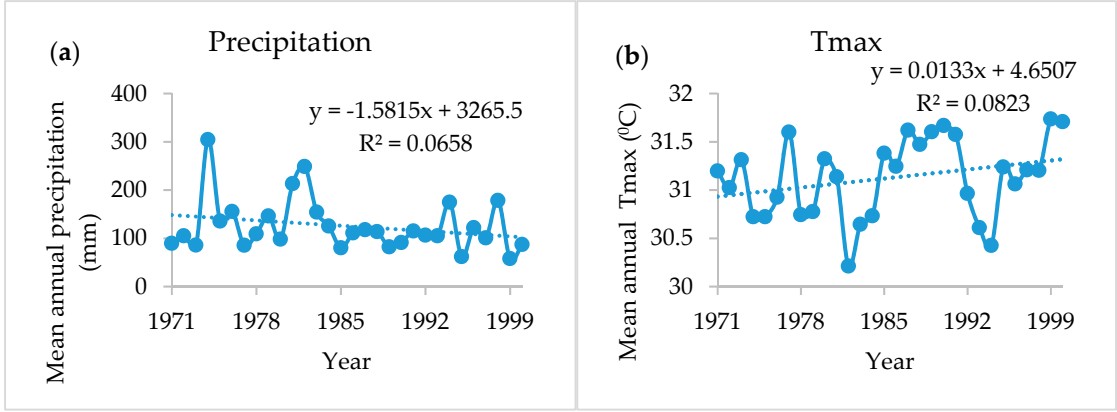

**Figure 5.** *Cont.*

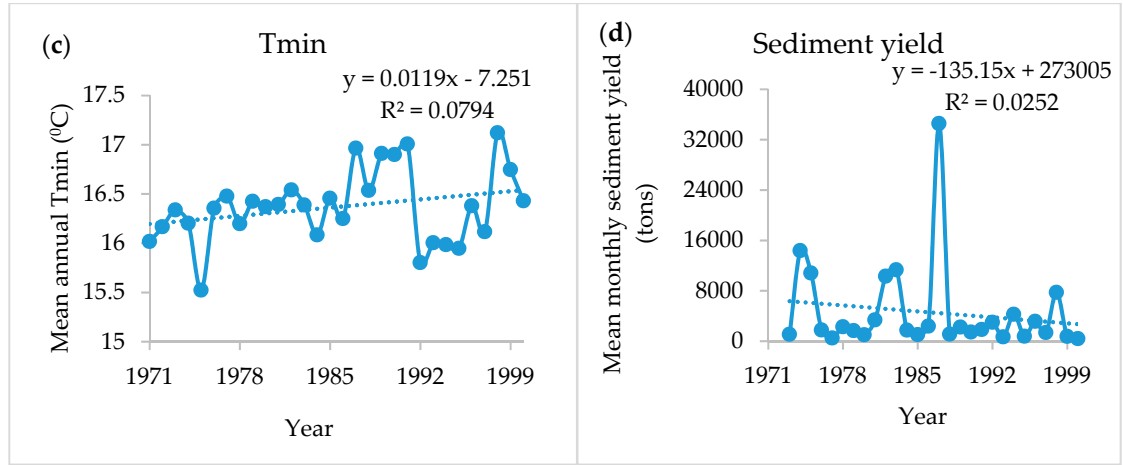

**Figure 5.** Historical trends of the annual precipitation (**a**), temperature maximum (Tmax) (**b**), temperature minimum (Tmin) (**c**), and sediment yield (**d**) from 1971–2000 in the Logiya watershed.

**Table 2.** Statistical summary of annual baseline data (1971–2000).

| Descriptive Statistics | Precipitation (mm) | Tmax (°C) | Tmin (°C) | Sediment Yield (tons) |
|---|---|---|---|---|
| Max | 304.5 | 31.7 | 17.1 | 34547.3 |
| Min | 58.0 | 30.2 | 15.5 | 381.6 |
| SD | 54.3 | 0.4 | 0.4 | 7001.4 |
| Mean | 125.4 | 31.1 | 16.4 | 4528.4 |
| CV (%) | 43.3 | 1.3 | 2.3 | 154.6 |
| *p*-value | 0.272 | 0.101 | 0.117 | 0.299 |
| Alpha | 0.05 | 0.05 | 0.05 | 0.05 |

SD: standard deviation, CV: coefficient of variance.

## 3.2. Evaluating Accuracy of Climate Model Simulations

Impact assessment of climate change necessitates evaluation of the climate model's accuracy. For this reason, in this study, the model's performance was evaluated using statistical measures such as bias, CV, and CC [54,85]. Observed mean annual precipitation as well as bias-corrected and bias-uncorrected historical RCPs framed for the Logiya watershed were 742.46, 733.65, and 408.86 mm, respectively (Table 3). Despite a slight underestimation in model simulation, the mean annual precipitation of the observed and the model output showed a good relationship. The results revealed the degree of rainfall coefficient variability (CV = 2.60%) for observed and bias-corrected RCP data (CV = 3.12%) and for bias-corrected RCP data, while there was slight underestimation with bias = −1.19%, which suggested that, in the watershed, the precipitation is well captured or represented, and a good linear relationship was established, as evaluated by correlation (CC = 0.56) for the bias-corrected model simulation data. The average result downscaled dynamically with maximum and minimum temperatures, which also virtually indicated a good relationship with the observed data. Therefore, comparisons of observed and corrected RCP data were consistent (Figure 6).

**Table 3.** Performance evaluation of observed and downscaled historical representative concentration pathway (RCP) datasets of the average annual precipitation in the Logiya watershed (1988–2016).

| | Mean Annual Precipitation (mm) | CV (%) | Bias (%) | Correlation |
|---|---|---|---|---|
| Observed | 742.46 | 2.60 | - | - |
| Bias-corrected RCP | 733.65 | 3.12 | −1.19 | 0.56 |
| Un-corrected RCP | 408.86 | 2.32 | −44.93 | 0.34 |

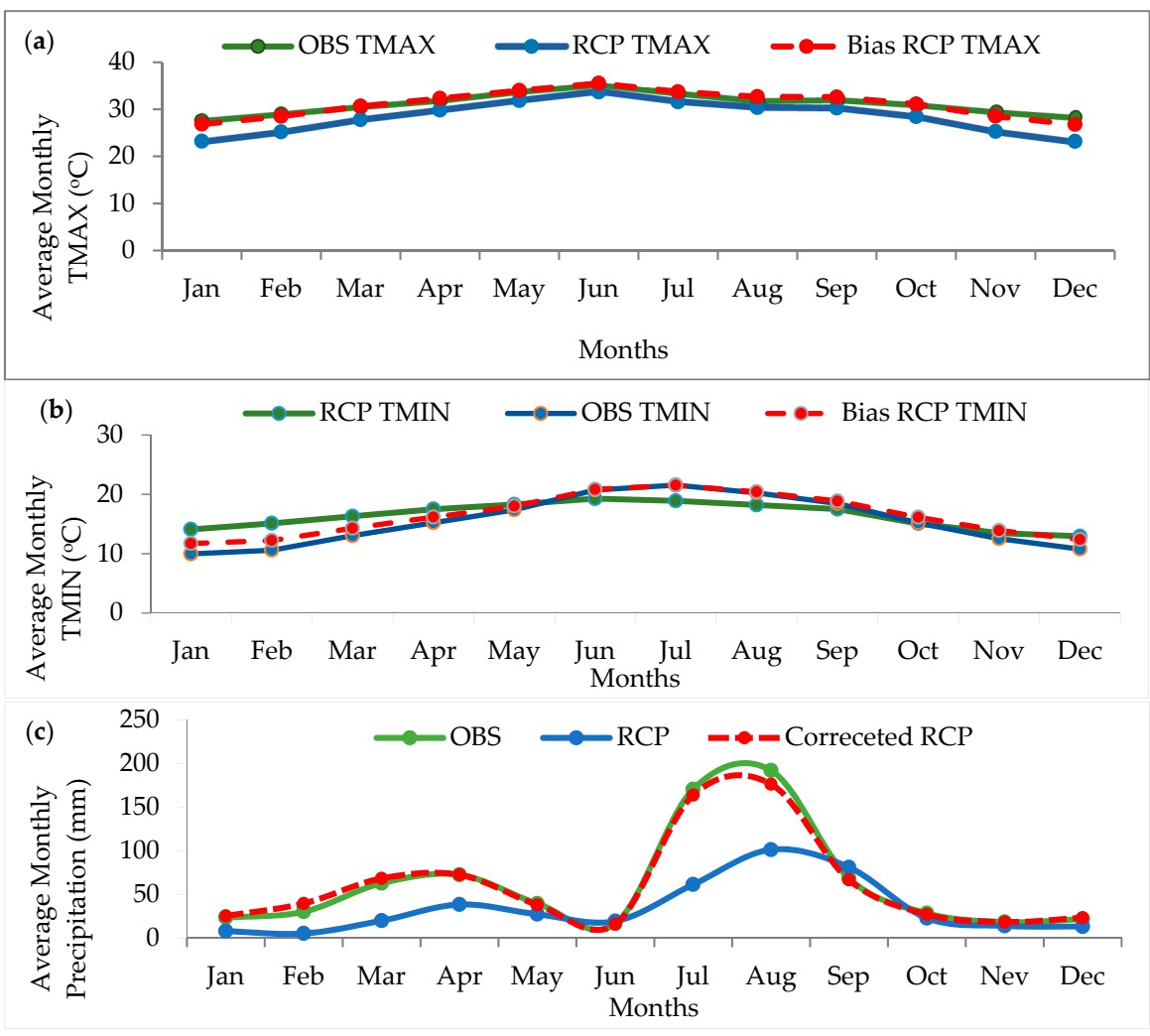

**Figure 6.** Observed and downscaled average monthly maximum temperature (**a**), minimum temperature (**b**), and precipitation (**c**) of observed vs. bias-corrected and un-corrected RCP time series (1988–2016) of the Logiya watershed.

### 3.3. Future Climate Projection

This study also analyzed future climate patterns such as precipitation, temperature, and evapotranspiration under RCP4.5 and RCP8.5 scenarios. The evaluation was made over two consecutive 30-years period of the 2030s and the 2060s, respectively. Accordingly, the future climate change was estimated for each RCP based on the baseline period (1971–2000). The mean annual precipitations were projected under RCP4.5 to increase by 0.9% and 4.11% for the 2030s and the 2060s, respectively. Likewise, for RCP8.5, the mean annual precipitation increases were found to be 3.55% in the 2030s and 8.69% in the 2060s. In both RCP4.5 and RCP8.5 scenarios, the results showed an increase in precipitation patterns in July–January, whereas they showed decreased precipitation patterns in February–June in the 2030s and the 2060s, respectively. Besides, the results pointed out that the alteration of precipitation patterns depended on the seasonality of the watershed. Supporting the present study's findings, Intergovernmental Panel on Climate Change (IPCC) [10] reported an increase of precipitation in rainy seasons with heavy precipitation events over the world. The global precipitation pattern is expected to increase up to 20% [10]. Getahun and Lanen [86] reported projected change in main rainy season precipitation with less than ±20% in all GCMs in the Upper Awash River Basin. The predicted increase of precipitation was unevenly distributed in a year, which would further increase the seasonal variation of precipitation in the watershed [19].

In this study, the predicted temperatures in all months might increase in both RCP scenarios. Under RCP4.5, changes in monthly maximum temperatures ranged from 0.1 °C to 0.47 °C with an average annual of 0.31 °C by the 2030s and from 0.07 °C to 0.26 °C with an average annual of 0.17 °C by the 2060s. For the RCP8.5 scenario, the monthly maximum temperature increased from 0.03 °C to 0.73 °C and from 0.09 °C to 0.27 °C with average annuals of 0.33 °C and 0.19 °C for the 2030s and the 2060s, respectively. The results also indicated that all the values of monthly minimum temperature increased from 0.03 °C to 0.45 °C by the 2030s and from 0.03 °C to 0.49 °C by the 2060s under RCP4.5 with average annuals of 0.25 °C and 0.32 °C, respectively, compared to the baseline period. Additionally, in the RCP8.5 scenario, the minimum temperature increased from 0.05 °C to 0.56 °C for the 2030s and from 0.02 °C to 0.57 °C for the 2060s. The average annual minimum temperatures were 0.28 °C and 0.35 °C under RCP8.5 for the 2030s and the 2060s, respectively.

These results agreed with the findings reported by Daba et al. [87] for the Upper Awash Sub-Basin showing an increase in predicted maximum and minimum temperatures, but the changes in temperature were uneven throughout the month. Getahun and Lanen [86] and Kinfe [88] projected maximum and minimum temperature increases in old SRES and in both RCP4.5 and RCP8.5 scenarios in the Upper Awash River Basin. Overall, these findings were in line with previous studies conducted in the Awash Basin [86–90]. The increased temperature was additionally related with an increase in evapotranspiration in all RCP scenarios. Finally, the projected increment in temperature change under RCP4.5 was lower than under RCP8.5 for both maximum and minimum temperatures because RCP8.5 had higher emission of greenhouse gases than the RCP4.5 scenario [10].

*3.4. SWAT Model Calibration and Validation*

3.4.1. Sensitivity Parameters of Streamflow Analysis

Twelve of the most sensitive parameters were selected for streamflow simulation of the Logiya watershed with their fitted value ranked in Table 4.

**Table 4.** Result of the sensitivity analysis parameters of streamflow [sensitivity parameters maximum, minimum, and fitted values using Sequential Uncertainty Fitting version-2 (SUFI-2)].

| Rank | Parameters | Descriptions | Fitted Value | Min | Max |
|------|-----------|--------------|--------------|-----|-----|
| 1 | ALPHA_BF | Base flow alpha factor (days) | 0.785 | 0 | 1 |
| 2 | CH_K2 | Effective hydraulic conductivity of the main channel | 77.49 | −0.05 | 500 |
| 3 | GW_DELAY | Groundwater delay (days) | 362.5 | 0 | 500 |
| 4 | CN-2 | SCS (soil conservation service) runoff curve number for moisture condition—II | 0.1 | −0.25 | +0.25 |
| 5 | REVAPMN | Threshold depth of water in the shallow aquifer for "revap" to occur (mm) | 477.01 | 0 | 500 |
| 6 | RCHRG-DP | Deep aquifer percolation fraction | 0.385 | 0 | 1 |
| 7 | SOL_K | Saturated hydraulic conductivity (mm/hr) | 1546.78 | 0 | 2000 |
| 8 | GWQMN | The threshold depth of water in shallow required for return flow (mm) | 2134.87 | 0 | 5000 |
| 9 | SURLAG | Surface runoff lag time | 16.89 | 0.05 | 24 |
| 10 | SOL_AWC | Soil available water capacity | 0.05 | 0 | 1 |
| 11 | EPCO | Soil evaporation compensation factor | 0.69 | 0 | 1 |
| 12 | SOL_ALB | Moist soil albedo | 0.1 | 0 | 0.25 |

3.4.2. Streamflow Calibration and Validation

Streamflow calibration and validation was done by comparing the observed and the simulated flow values for a 16-years period (1990–2005). The first two years (1990–1991) were considered as a warm-up period, the 1992–1999 period was used for the model calibration, and the 2000–2005 period was used for the validation period. Both calibration and validation for streamflow simulation obtained satisfactory results of fit with $R^2$ of 0.8 and 0.77 and NSE coefficients of 0.73 and 0.63, respectively (Table 5 and Figure 7). The results revealed that the variation pattern of simulated streamflow was generally consistent with that of the observed streamflow.

**Table 5.** Streamflow calibration and validation results (monthly) at the Logiya outlet gauging station

| Variable | Calibration | Validation |
|----------|-------------|------------|
| $R^2$ | 0.8 | 0.77 |
| NSE | 0.73 | 0.63 |
| PBIAS | +30.5 | +38.1 |

NSE: Nash–Sutcliffe efficiency; PBIAS: percent bias.

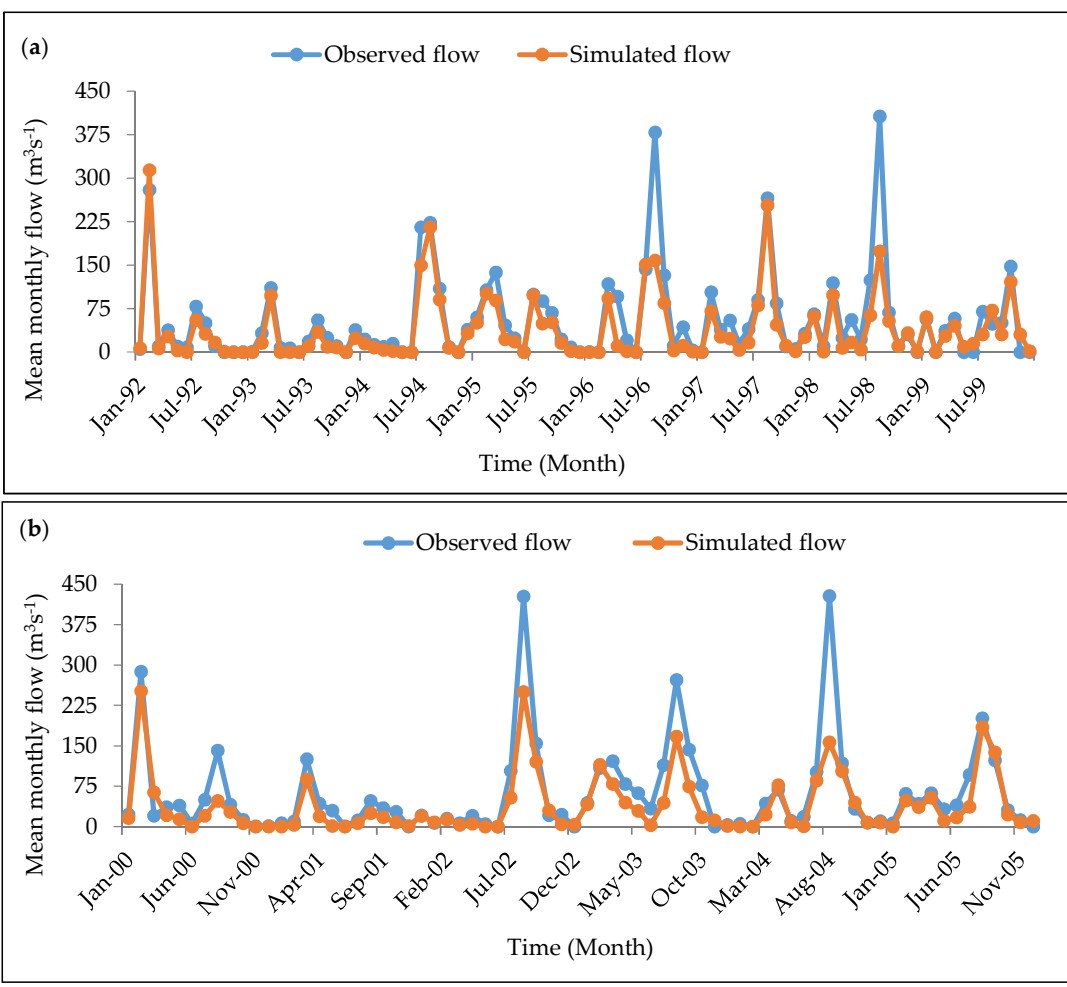

**Figure 7.** Soil and Water Assessment Tool (SWAT) model calibration (**a**) and validation (**b**) for streamflow at the monthly time scale during 1992–1999 and 2000–2005, respectively.

### 3.4.3. Sediment Yield Calibration and Validation

The most sensitive parameters that affect sediment yields used in the Logiya watershed are the linear parameter for calculating the maximum amount of sediment (SPCON), the USLE equation support particle factor (USLE_P), and the exponent parameter for calculating sediment re-entrained channel sediment routing (SPEXP) (Table 6).

**Table 6.** Sensitivity analysis parameters of Sediment yield with fitted values using SUFI-2.

| Rank | Parameters | Descriptions | Fitted Value | Min | Max |
|------|-----------|--------------|--------------|-----|-----|
| 1 | SPCON | Linear factor for channel sediment routing | 0.005 | 0.0001 | 0.01 |
| 2 | SPEXP | Exponential factor for channel sediment routing | 1.387 | 1 | 2 |
| 3 | USLE-P | USLE support Practice factor | 0.275 | 0 | 1 |

SPCON: maximum amount of sediment; SPEXP: sediment re-entrained channel sediment routing; USLE-P: universal soil loss equation support particle factor.

The SWAT-CUP in parallel gives the best simulation of sediment yield using the most sensitive parameter values. Using fitted parameters of calibration, the simulation was undertaken for the validation period, and its performance was checked with the observed data. The best simulation performances of sediment yield results using these fitted parameters and the values for monthly calibration and validation are shown in Table 7 and Figure 8.

**Table 7.** Monthly sediment yield calibration and validation results at the Logiya outlet gauging

| Variable | Calibration | Validation |
|:---:|:---:|:---:|
| $R^2$ | 0.83 | 0.85 |
| NSE | 0.79 | 0.76 |
| PBIAS | −23.4 | −25.0 |

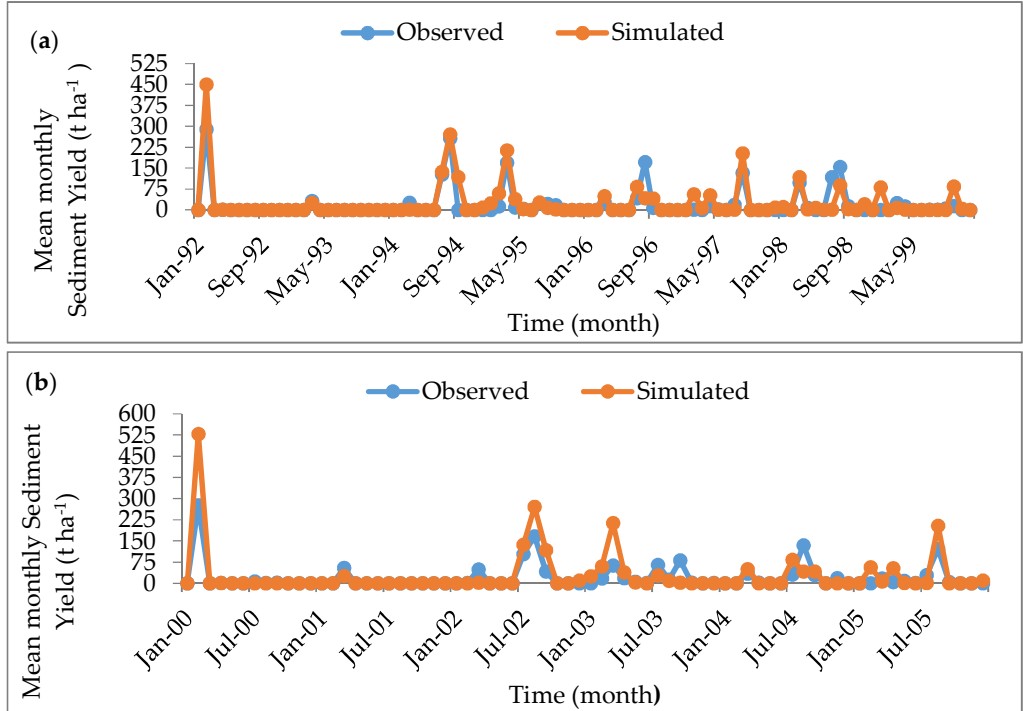

**Figure 8.** SWAT model calibration (**a**) and validation (**b**) for sediment yield at the monthly time scale during 1992–1999 and 2000–2005 respectively.

Numerous studies (e.g., Setegn et al. [8], Santhi et al. [81], Moriasi et al. [82], Benaman et al. [91]) suggested that the prediction efficiency of the calibrated model can be a good agreement if $R^2$ and NSE values are greater than 0.6 and when the value of PBIAS is between $\pm15 \leq$ and $\leq\pm30$. Since, in the present study, the values of $R^2$ and NSE were 0.83 and 0.79, respectively, the calibration and the validation of the observed and the simulated streamflow and sediment yield of the Logiya watershed showed good relations with reasonable accuracy [8,81,82,90].

The results indicated that, for the Logiya watershed, the SWAT model simulated streamflow and sediment yield with reasonable accuracy. Mean annual sediment loads of observed and simulated values in the years 1992–2005 were 19.53 tons ha$^{-1}$ and 23.66 tons ha$^{-1}$, respectively. This indicated the estimates of sediment load in the model were greater than the observed values. This may have been due to limited sediment data, quality and constraint of weather data, streamflow, errors during data recording, and uneven distribution of gauging stations in the watershed [75,92,93]. Underestimates and overestimates of mean monthly data have been recorded in the model outputs over the observed high flow months such as February, March, April, July, August, and September (Figure 9).

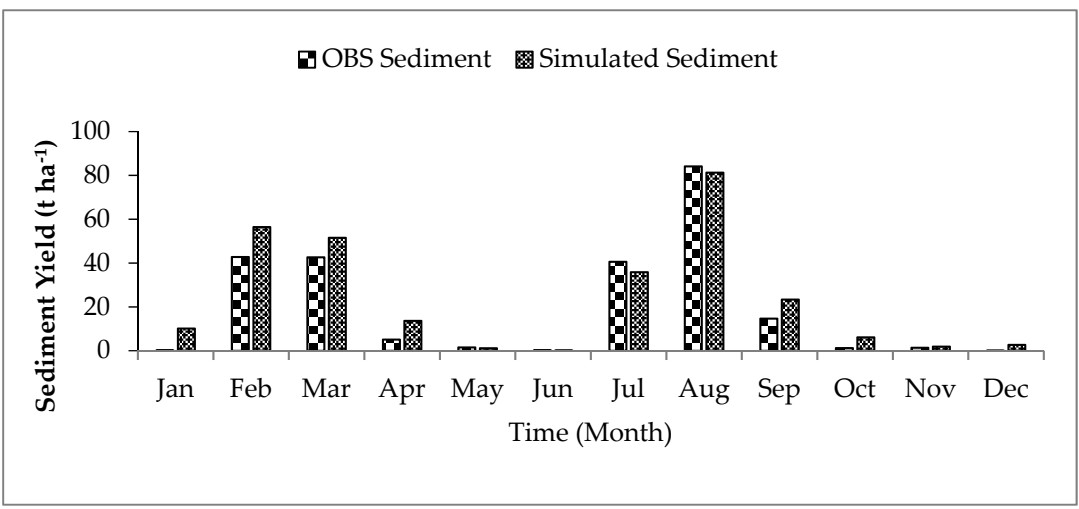

**Figure 9.** Mean monthly observed and simulated sediment load (1992–2005).

*3.5. Spatial Variation of Sediment Yield under Different Climate Scenarios*

It was found that the SWAT model is advantageous in providing quantitative estimates of soil loss and sediment yield and their spatial variations in different sub-watershed by considering HRU [94–96]. This result plays a critical role for the detailed understanding and evaluation of the amount of sediment yield as well as for designing an appropriate management scenario at each sub-watershed. The spatially distributed sediment yield in each sub-watershed under different climate scenarios is demonstrated in Figures 10 and 11. During the observed periods, the rate of sediment yield ranged from 3.28 t ha$^{-1}$ yr$^{-1}$ to 51.77 t ha$^{-1}$ yr$^{-1}$, which corresponded to the sub-watersheds 30 and 27 of the Logiya watershed, respectively, with a mean value of 19.07 t ha$^{-1}$ yr$^{-1}$.

Besides, the projected sediment yield estimated under both RCP4.5 and RCP8.5 scenarios showed the spatial variation across the watershed. Severely eroded areas were situated at downstream parts of the study area. These findings agreed with studies conducted on the Lower Awash basin and the Upper Blue Nile River basin [37,94]. For RCP4.5, by the 2030s and the 2060s, the mean values of sediment yields were 2.57 t ha$^{-1}$ yr$^{-1}$ and 2.80 t ha$^{-1}$ yr$^{-1}$, respectively. Also, under the RCP8.5 scenario, the mean values of sediment yields were 3.31 t ha$^{-1}$ yr$^{-1}$ and 4.07 t ha$^{-1}$ yr$^{-1}$ for the 2030s and the 2060s, respectively. Results indicated that for RCP4.5 and RCP8.5, the maximum and the minimum sediment loads of the watershed occurred in the sub-watersheds 27 and 19, respectively, in both time series data. In the case of the Logiya watershed, more than two-thirds (76% of the total study area) of the observed soil loss rate was above the tolerable erosion limit suggested by Hurni [97] for various agro-climatic zones of Ethiopia, which ranges from 2 to 18 t ha$^{-1}$ yr$^{-1}$.

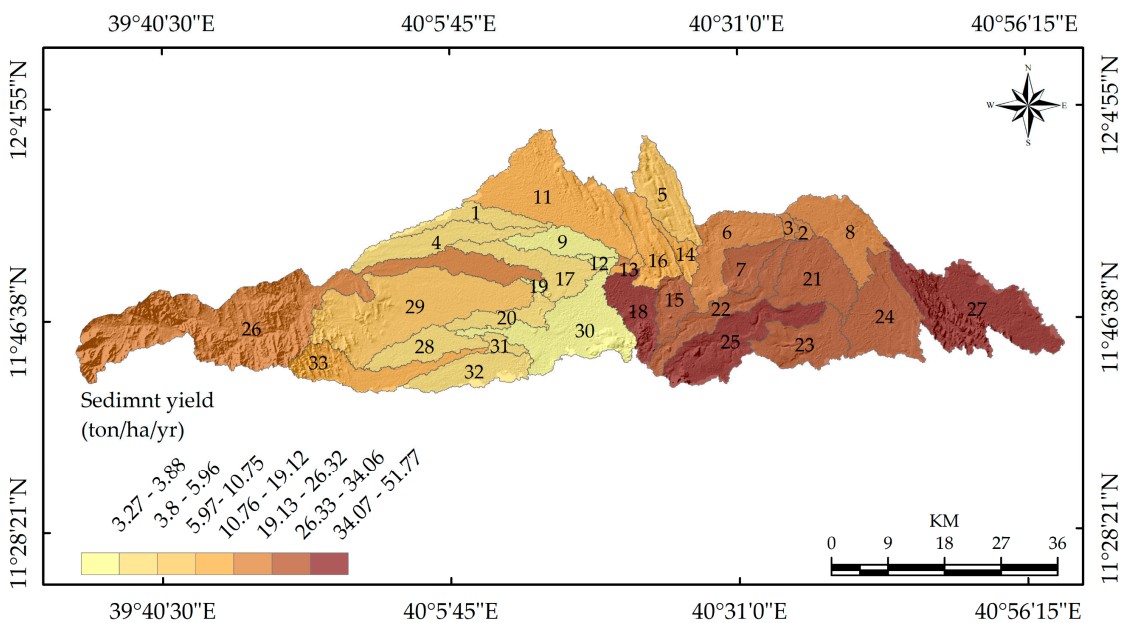

**Figure 10.** Spatial map of the observed sediment yield.

In addition, the results indicate that more severe erosions occurred in the downstream parts of the watershed, than in the upstream areas of the watershed. This shows that soil erosion was a serious phenomenon in the different sub-basins in this study area during observed and projected time series. During the observed and the projected future RCP scenarios, the maximum sediment load occurred in sub-basin 27 of the watershed. Also, bare land and Eutric Regosols were the most dominant LULC and soil type of sub-basin 27, respectively. All this indicate that LULC was more sensitive to soil erosion in this sub-basin. What's more, it can be understood that Eutric Regosols soil type mostly occurs in desert regions, and it occupies the main LULC of that area. Kefyalew [37] indicated that the Lower Awash basin was highly affected by flood and soil erosion year-to-year.

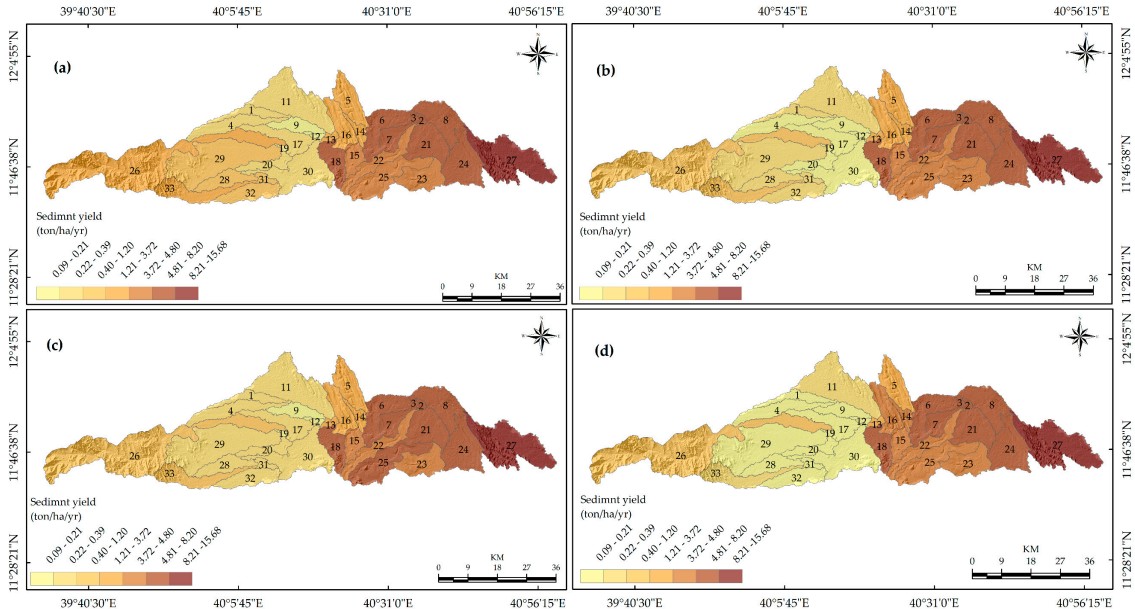

**Figure 11.** Spatial map of the annual sediment yield (**a**) in 2030s RCP4.5 (**b**) in 2030s RCP8.5 (**c**) in 2060s RCP4.5 and (**d**) in 2060s RCP8.5 in the Logiya watershed.

### 3.6. Impact of Climate Change on Sediment Yield and Streamflow in Logiya Watershed

Similar to future precipitation, sediment yield estimates showed an increasing pattern in July–January and a decreasing pattern in February–June in both RCP scenarios (Figure 12). The mean annual sediment yields were expected to increase by 4.42% and 8.08% in the 2030s and the 2060s, respectively, for RCP4.5. They were also expected to increase by 7.19% for the 2030s and the 10.79% for 2060s under the RCP8.5 scenario. The results indicated that precipitation and sediment yield had strong relationships, and this agreed with previous findings [12,23,98,99]. Moreover, the relationship between streamflow and sediment load at the outlet of the Logiya watershed was strongly related. Both streamflow and sediment yield were predicted to increase under both RCP scenarios. Mean annual streamflows were predicted to increase by 1.43% and 3.47% for RCP4.5 in the 2030s and the 2060s, respectively. For the RCP8.5 scenario, the model showed an increase of 2.81% by the 2030s and 5.48% by the 2060s. This implied that sediment yield at the outlet of the watershed was influenced by streamflow. Overall, both RCP scenarios showed fairly similar patterns of increment in projected monthly precipitation, streamflow, and sediment yield in a fair amount in the study area. Similar results were reported by Shrestha et al. [13], Adem et al. [28], and Azari et al. [5], all of whom showed increasing streamflows proportional to increasing sediment yield. Figure 12 displays the projected mean monthly sediment yield at the outlet of the Logiya watershed under all RCP scenarios.

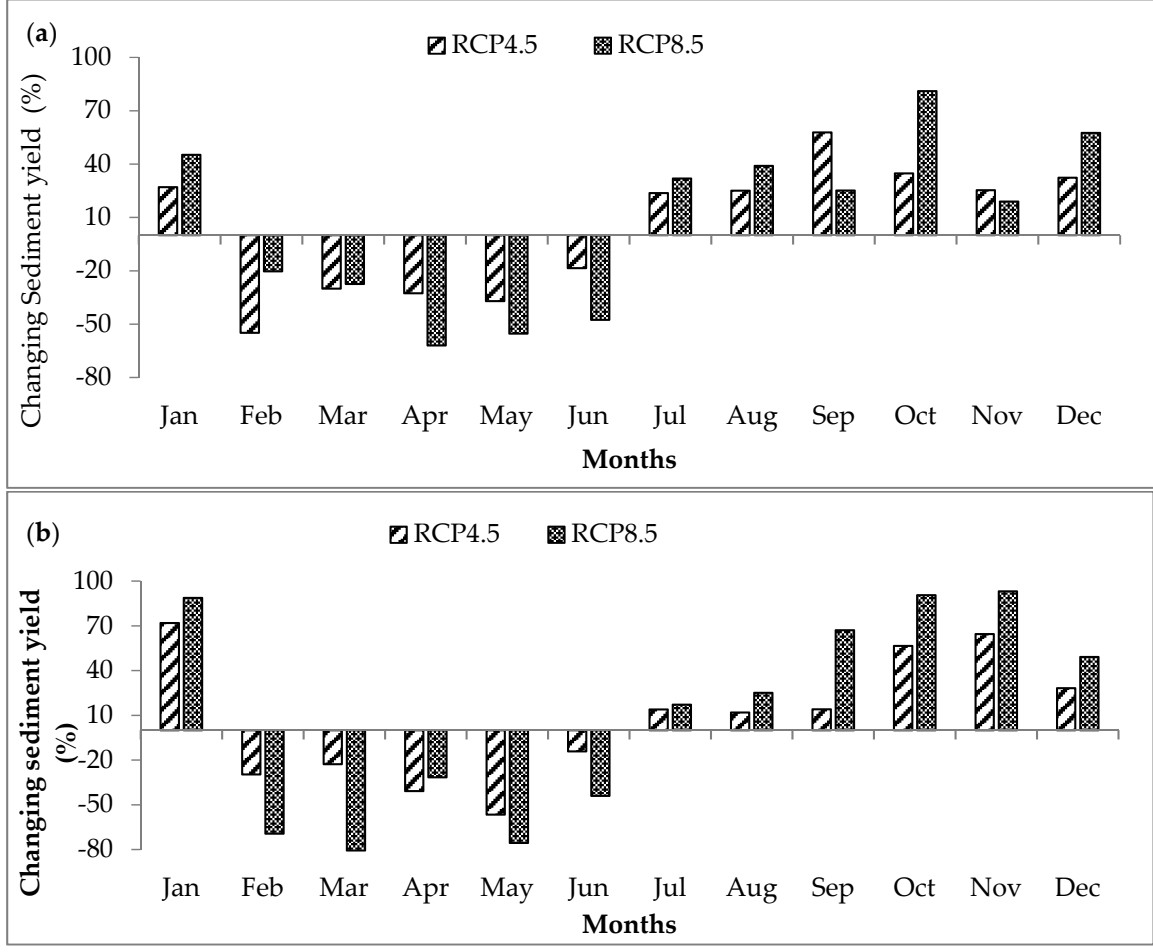

**Figure 12.** Projected change of mean monthly sediment yield (**a**) in the 2030s (**b**) in the 2060s from the baseline period in the Logiya watershed.

Figure 13 depicts the mean annual sediment load for RCP4.5 which was 23.51 tons ha$^{-1}$ and 22.82 tons ha$^{-1}$ for the 2030s and the 2060s, respectively. Also, for the RCP8.5 scenario, they were

24.3 tons ha$^{-1}$ by 2030s and 23.1 tons ha$^{-1}$ by the 2060s. The findings indicated that the monthly maximum and minimum sediment loads would happen in August−June. This pattern showed that the observed and projected sediment load in this study area were strongly related.

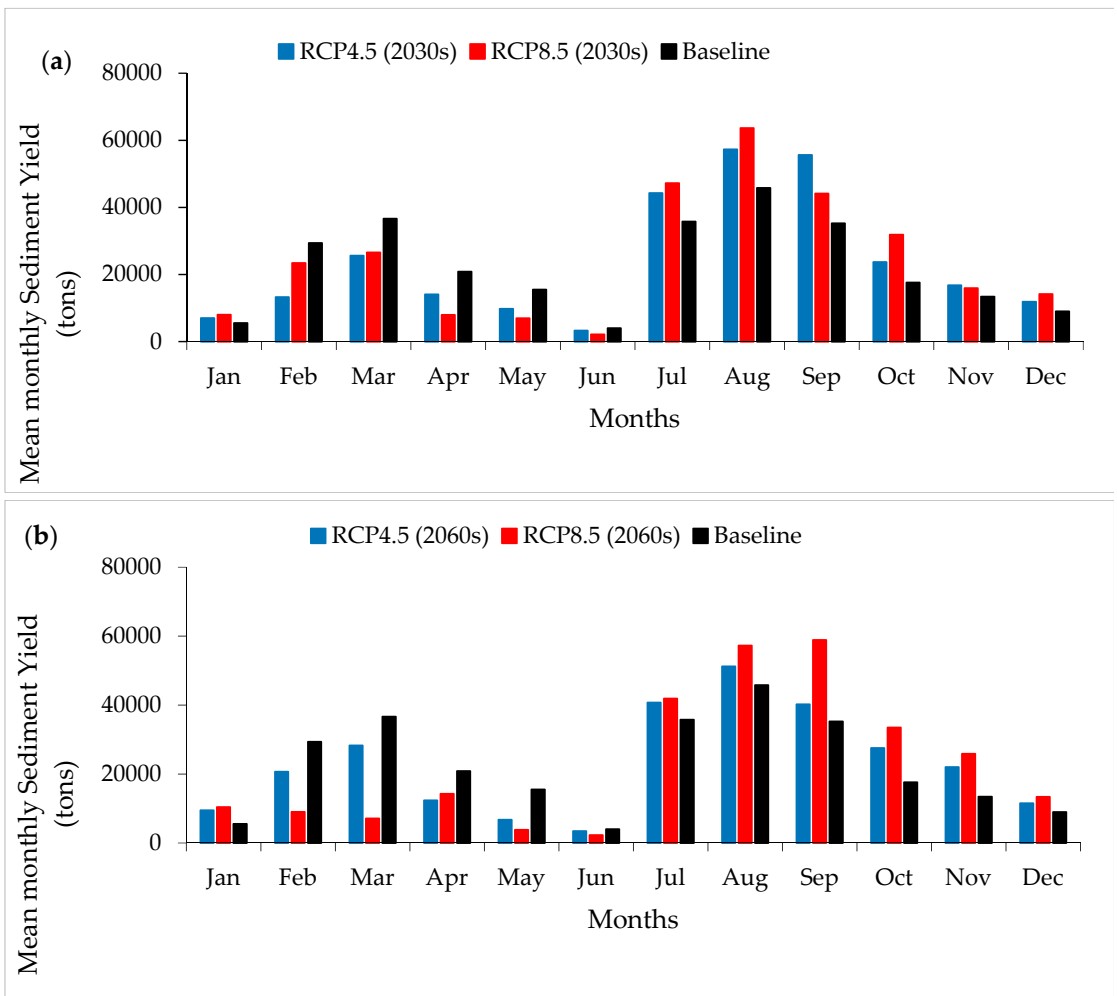

**Figure 13.** Mean monthly sediment yield in the 2030s (**a**) and the 2060s (**b**) projected at the outlet of the Logiya watershed.

## 4. Conclusions

The present study investigated the impacts of climate change on sediment yield from the Logiya watershed in the Lower Awash Basin, Ethiopia. RCMs in CORDEX-Africa derived from the HadGEM2-ES Global Climate Model were used for RCP4.5 and RCP8.5 by the 2030s and by the 2060s as compared to the baseline period (1971–2000). The SWAT hydrological model was used to simulate present and future changes in sediment yield, and the SUFI-2 algorithm in the SWAT-CUP program was used for parameter adjustment. Calibration, validation, and uncertainty analyses for sediment suggested that the SWAT model could be applied to simulate future changes in sediment yields due to ultimate climate change.

Results indicated that mean annual precipitation as well as maximum and minimum temperatures would increase under RCP4.5 and RCP8.5 scenarios. The projected annual precipitations of RCP4.5 in the 2030s and the 2060s would increase by 0.9% and 4.11%, respectively, and in the RCP8.5 scenario would increase from 3.55% and 8.69% for the 2030s and the 2060s, respectively. The RCP8.5 scenario was changed by a higher magnitude than the RCP4.5 scenario because RCP8.5 had a higher greenhouse gas emission scenario with a higher degree of global warming than the RCP4.5 scenario.

Results of climate change impacts on sediment yield clearly showed that sediment loss would increase in all RCP scenarios. The mean annual sediment yields were predicted to increase by 4.42% by the 2030s and by 8.08% by the 2060s under RCP4.5. For RCP8.5, they were found to be 7.19% and 10.79% by the 2030s and the 2060s, respectively. Moreover, the projected sediment yield was found to be a similar pattern of precipitation and streamflow. This indicated that precipitation, streamflow, and sediment load would operate in good relation with this watershed. Finally, the results showed that the amount of sediment load would vary in each sub-watershed due to changes in precipitation and temperature in the 2030s and the 2060s for both RCP4.5 and RCP8.5 scenarios. The maximum and the minimum sediment yields would occur in sub-basins 27 and 19 in the watershed.

**Author Contributions:** All authors made substantial contribution to the development of this manuscript. N.B.J. was in charge of conceptualization, data analysis and writing of original draft. B.G., A.E.H., G.W.W. and F.B. reviewed, edited and improved the manuscript. All authors read and approved the final manuscript.

**Funding:** This research received no external funding.

**Conflicts of Interest:** The authors declare no conflict of interest.

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
