# Peer review of "Evaluation of the Impacts of Climate Change on Sediment Yield from the Logiya Watershed, Lower Awash Basin, Ethiopia"

_hydrology, doi:10.3390/hydrology6030081_

Round 1

Reviewer 1 Report

The manuscript number hydrology-549106, entitled Evaluation of Climate Change Impact on Sediment Yield from Logiya Watershed, Lower Awash Basin, Ethiopia, aims to assess the impacts of climate change on sediment yield from the Logiya Watershed, which is a sub-basin of the Lower Awash Basin in Ethiopia. The authors used two different future scenarios, representative of RCP4.5 and RCP8.5 concentration pathways of IPCC, to force the SWAT model. Results have been evaluated in comparison to those obtained with a reference scenario (baseline). The model was first calibrated and validated for two different periods and performance evaluated by means of different statistical indexes (R2, NSE, and PBIAS).

After carefully reading the manuscript, my biggest concern is about the novelty and the soundness of the work. The manuscript seems to be just the umpteenth application of an already known model (SWAT) and already used framework (a baseline plus future scenarios from IPCC report) to evaluate and compare the hydrological and sedimentological responses of a basin to those different scenarios. Moreover, as highlighted in some studies (see as an example Francipane, A., S. Fatichi, V.Y. Ivanov, and L.V. Noto (2015), Stochastic assessment of climate impacts on hydrology and geomorphology of semiarid headwater basins using a physically based model, J. Geophys. Res. Earth Surf., 120, 507–533, doi:10.1002/2014JF003232), when working with climate projections there is always a stochastic variability to take into account that moves to hydrological and sedimentological responses. In other words, the climate change effects on sediment transport cannot be properly addressed without understanding of stochastic variability representative of historic conditions as well as climate model and stochastic uncertainties associated with projected future conditions. That said, I think there are several issues that, if addressed, would make the paper stronger. I recommend publication contingent on major revisions as described in the following.

You can find my general comments in the following, while other comments, revisions, and suggestions are in the attached pdf file.

General comments

The introduction is very poor in content. There are few references to past works and often are too old works or to the Ethiopian case. The authors did not report the state of the art in the field of erosion and impact of climate changes on that, which has been very well studied over the world in the past.

In Section 2.2.3 there is much confusion in the description of future climate scenarios. It is not clear as the authors generated those. I think the section should be reorganized in a simpler and more organic way. This would help in understanding also the experiments and their results.

Although I am not a native English speaker, I feel that the meaning and clarity of the manuscript will be helped by a careful proofread, preferably by a native English speaker. Authors can find some indications in the pdf file.

About Figures

There is something strange in Figure 1. The Digital Elevation Model (DEM) has a strange salt and pepper effect that suggests it is affected by noise. Please double check your DEM.

Also, I think the authors should strive to describe better the content of Figure 1 within the manuscript (e.g., meteorological and flow stations).

Figure 3 is too small. The text in the figure is not readable and the content of figures is not valuable.

Many figures are not called within the manuscript (e.g., 2, 3, 5, 10, 12).

Other comments

You can find other comments on the pdf file of the manuscript in the form of tracked changes.

Author Response

Reviewer #1:

General comments:

After carefully reading the manuscript, my biggest concern is about the novelty and the soundness of the work. The manuscript seems to be just the umpteenth application of an already known model (SWAT) and already used framework (a baseline plus future scenarios from IPCC report) to evaluate and compare the hydrological and sedimentological responses of a basin to those different scenarios. Moreover, as highlighted in some studies (see as an example Francipane, A., S. Fatichi, V.Y. Ivanov, and L.V. Noto (2015), Stochastic assessment of climate impacts on hydrology and geomorphology of semiarid headwater basins using a physically based model, J. Geophys. Res. Earth Surf., 120, 507–533, doi:10.1002/2014JF003232), when working with climate projections there is always a stochastic variability to take into account that moves to hydrological and sedimentological responses. In other words, the climate change effects on sediment transport cannot be properly addressed without understanding of stochastic variability representative of historic conditions as well as climate model and stochastic uncertainties associated with projected future conditions. That said, I think there are several issues that, if addressed, would make the paper stronger.

[Response 1]: Many thanks to the reviewer 1 for his/her constructive comments and providing such detailed and helpful comments, which helped to enrich the quality of the manuscript. As per the comments, we have added new subtopics and discussed the trend of historical climate variability. We used statistical variability such as maximum values, minimum values, SD, mean and CV of precipitation, maximum and minimum temperature, and sediment yield.

The introduction is very poor in content. There are few references to past works and often are too old works or to the Ethiopian case. The authors did not report the state of the art in the field of erosion and impact of climate changes on that, which has been very well studied over the world in the past.

[Response 2]: We thank the reviewer for his/her genuine comment, the introduction section is extensively edited in the current version of the manuscript. We have now added discussion on the impact of climate change on soil erosion with inclusion of new references from recent peer reviewed journals from different parts of the World.

In Section 2.2.3 there is much confusion in the description of future climate scenarios. It is not clear as the authors generated those. I think the section should be reorganized in a simpler and more organic way. This would help in understanding also the experiments and their results.

  [Response 3]: Apologise for the confusions caused. The paragraph has been revised to improve clarity on methods (Line 193-210).

Although I am not a native English speaker, I feel that the meaning and clarity of the manuscript will be helped by a careful proofread, preferably by a native English speaker. Authors can find some indications in the pdf file.

 [Response 4]: The attached document ‘peer-review-4664016.v1.pdf’ is checked and all the comments were incorporated in the revised version of the manuscript. The manuscript is edited again to improve its grammar.

Specific comments:

There are more recent studies. Please have a look at this interesting and recent study as well: Francipane, A., S. Fatichi, V. Y. Ivanov, and L. V. Noto (2015), Stochastic assessment of climate impacts on hydrology and geomorphology of semiarid headwater basins using a physically based model, J. Geophys. Res. Earth Surf., 120, 507–533, doi:10.1002/2014JF003232.

     [Response 1] We thank the reviewer for his/her suggestion, we have cited the study by Francipane et al. (2015).

 between the study periods and related those areas of change with the soil erosion risk (Line: 165-185).

Most of references is referred to Ethiopia. At this stage, the authors are discussing about the phenomenon of climate changes and its consequences in general terms. I think the authors should have a look at more recent articles and also at what happens in different parts of the world. Pleas have a starting point from the followings: Viola, F., Francipane, A., Caracciolo, D., Pumo, D., La Loggia, G., Noto, L.V., 2016. Coevolution of hydrological components under climate change scenarios in Mediterranean area. Sci. Total Environ. 544, 515–524. http://dx.doi.org/10.1016/j.scitoten.2015.12.004. Pumo, D., Caracciolo, D., Viola, F., & Noto, L. V. (2016). Climate change effects on the hydrological regime of small non‐perennial river basins. Science of the Total Environment, 542 (Part A, 76–92. https://doi.org/10.1016/j.scitoten.2015.10.109. Arnone E, Pumo D, Francipane A, La Loggia G, Noto LV. The role of urban growth, climate change, and their interplay in altering runoff extremes. Hydrological Processes. 2018; 32:1755–1770. https://doi.org/10.1002/hyp.13141.

[Response 2]: Many thanks for pointing this out. We referred the suggested journals and other similar work from different parts of the world.

Flood (s)

[Response 3]: The typo has now been corrected.

its

[Response 4]: The comment is accepted. The sentence has now been rewritten in the current version of the manuscript (Line 79-81).

3,151.86 km2

[Response 5]: The typo has now been corrected throughout the manuscript.

named:

[Response 6]: The word has now added.

There is something strange in Figure 1. I mean, the Digital Elevation Model (DEM) has a strange salt and pepper effect that suggests it is affected by noise. Please double check your DEM. Also, I think the authors should strive to describe better the content of the Figure 1 in the manuscript (e.g., meteorological and flow stations). The text in the figure is not readable and the content of figures is not valuable.

[Response 7]: Apologise for the confusions caused. We resketched the figure to better enhance the visibility of the content in Figure 1.

Many figures are not called within the manuscript (e.g., 2, 3, 5, 10, 12)

[Response 8]: All figures have now recalled and described.

It seems something is missing here.

[Response 9]: The sentence has rewritten to improve the clarity (Line 135-137)

How did the authors fill the gap in data? IT is not clear.

[Response 10]: Thanks for the suggestion. The methods have now addressed (Line 115-158)

Authors should present Figure 2a firstly and then the Figure 2b. Please change their order. Also, I think they meant Figure 3.

[Response 11]: The comment is accepted. The order of figures was rearranged.

I would move this part in the previous section 2.2.3.

[Response 12]: The sentence has now been moved to 2.2.3. Soil Data.

It is not clear what the authors mean. Do they mean between 1951 and 2005 as historical scenario and between 2006 and 2100 as projected scenario? Please clarify.

[Response 13]: Sorry, for the confusion made. The sentence has now rewritten for better clarification. The period between 1951 and 2005 was taken as historical scenario and between 2006 and 2100 as projected scenario.

and

[Response 14]: The typo has now been corrected.

Why did the authors use a power function? Did they try to use a linear regression?

 [Response 15]: The power function (instead of a linear regression) was used to better illustrate the nonlinearity between sediment and stream flow.

Agricultural

 [Response 16]: The typo has now been corrected.

and

 [Response 17]: The typo has now been corrected.

Please define now what HRU stands for.

 [Response 18]: The full names of the acronyms have now described first and used consistently throughout the manuscript.

In what is it modified? Please add at least a reference.

 [Response 19]: We have now cited the reference of the method.

service (system)

[Response 20]: The typo has now been corrected.

They are all subscripts.

[Response 21]: Comment is accepted and corrected.

allowed (allows the user)

 [Response 22]: The typo has now been corrected.

concept (s)

 [Response 23]: The typo has now been corrected.

were performed

[Response 24]: The grammatical error is rectified.

This should be Table 2.

[Response 25]: The error is now corrected.

Mean

[Response 26]: The mean has now been added.

Please use an ascending order in Figures (i.e., a, b, c).

[Response 27]: The figures name was rearranged in an ascending order.

were

[Response 28]: The grammatical error is corrected.

and

[Response 29]: The PBIAS is replaced by and.

The performance rating of observed and simulated sediment is good

 [Response 30]: The grammatical errors have now been corrected.

since

 [Response 31]: The comment is accepted, therefore is replaced by since.

please add a blank space.

[Response 32]: The blank space has now added.

Move this before the Figure 11. You cannot present Figure 11 before to recall it, at least once, in the manuscript.

[Response 33]: The sentence has moved before Figure 11 (renamed as Figure 12 in the current version of the manuscript).

Reviewer 2 Report

Overall, it is a good and interesting paper, well written and structured, which has clear objectives. The exploration of future impacts of climate change on water resources, sediments yields in particular in this article, has gained significant attention and is a topic of wide interest. The methodology and model used are well described and the results produced are reasonable and are justified by the data and methods used. I recommend the authors to apply some minor revisions to improve the comprehensiveness and readability of their paper.

1)  In Table 5, correct SPEXY.

2) In Figure 10, the classes should remain unchanged between scenarios in order to show clearly what differs in space each time.

3) This is the case for Figure 11. Consider to keep the range of values in the y axis the same between (a) and (b). Move month labels down, not to overlap with the bars.

4) I think that the 'Discussion' is rather poor and should be extended a bit. I would recommend to move the largest part of the 'Conclusions' section to the Discussion and leave a very small and concise paragraph as 'Conclusions'.

Author Response

Reviewer #2:

General comments:

Overall, it is a good and interesting paper, well written and structured, which has clear objectives. The exploration of future impacts of climate change on water resources, sediments yields in particular in this article, has gained significant attention and is a topic of wide interest. The methodology and model used are well described and the results produced are reasonable and are justified by the data and methods used. I recommend the authors to apply some minor revisions to improve the comprehensiveness and readability of their paper.

 [Response 1]: Many thanks to the Reviewer 2 for the thoughtful comments and constructive suggestions, which helped to enhance the quality of the manuscript. The manuscript is edited again to improve its the comprehensiveness and readability. We have addressed all the comments from the anonymous review. Please see our detailed responses below.

Specific comments

In Table 5, correct SPEXP

[Response 2]: Thanks for your comment. The typo has now been corrected

In Figure 10, the classes should remain unchanged between scenarios in order to show clearly what differs in space each time.

[Response 3]: The range of classes value has made uniform.

This is the case for Figure 11. Consider to keep the range of values in the y axis the same between (a) and (b). Move month labels down, not to overlap with the bars.

[Response 4]: The figure is resketched to correct the overlaps and range of the values in the y axis has now been made uniform.

I think that the 'Discussion' is rather poor and should be extended a bit. I would recommend to move the largest part of the 'Conclusions' section to the Discussion and leave a very small and concise paragraph as 'Conclusions'.

[Response 5]: Thanks for the suggestion. The conclusions and discussion sections have been revised in the current version of the manuscript.

Round 2

Reviewer 1 Report

I thank the authors for their deep review. Despite I'm still not sure about some points (e.g., statistical explaining), I think the paper is more readable, robust, and complete now. I suggest its publication after the following minor revisions.

Line 673 (about Sediment Rating curve). I am still not sure about the power function. Even looking at the equation 6, which explains the relationship between the sediment flow and the streamflow, it is clear it is a linear relationship.

Line 822. It is not "Stastical" but "Statistical" and it is not "histrical" but "historical".

Line 892. It is not "ptternes" but "patterns".

Line 1187. It is nor "representes" but "represents".

Line 1441. Reference #18 is not Domenico C., Francesco A., but Caracciolo D., Viola F.

Author Response

Reviewer #1:

Specific comments:

I thank the authors for their deep review. Despite I'm still not sure about some points (e.g., statistical explaining), I think the paper is more readable, robust, and complete now. I suggest its publication after the following minor revisions.

[Response 1]: Many thanks to the Reviewer for the genuine and scholarly thoughtful comments and constructive suggestions, which helped to enhance the quality of the manuscript. We have now added about statistical explanations. Please see our detailed responses under 2.2.8 & 3.1. sub-sections in the manuscript.

Line 673 (about Sediment Rating curve). I am still not sure about the power function. Even looking at the equation 6, which explains the relationship between the sediment flow and the streamflow, it is clear it is a linear relationship.

[Response 2]: Sorry, for the confusion and the error we made. The Comment of Reviewer is accepted and it is corrected in the current version of the manuscript.

Line 822. It is not "Stastical" but "Statistical" and it is not "histrical" but "historical".

[Response 3]: The typos have now been corrected.

Line 892. It is not "ptternes" but "patterns".

[Response 4]: The typo has now been corrected.

Line 1187. It is nor "representes" but "represents".

[Response 5]: The typo has now been corrected.

Line 1441. Reference #17 is not Domenico C., Francesco A., but Caracciolo D., Viola F.

[Response 6]: Thanks for the suggestion. The citation error has now been corrected as per the journal reference style.
